# Deterioration Patterns in Patients Admitted for Severe COPD Exacerbation

**DOI:** 10.3390/diseases12110283

**Published:** 2024-11-07

**Authors:** Cristhian Alonso Correa-Gutiérrez, Zichen Ji, Irene Milagros Domínguez-Zabaleta, Javier Plaza-Hoz, Ion Gorrochategui-Mendigain, Ana López-de-Andrés, Rodrigo Jiménez-García, José Javier Zamorano-León, Luis Puente-Maestu, Javier de Miguel-Díez

**Affiliations:** 1Respiratory Department, Gregorio Marañón General University Hospital, 28007 Madrid, Spain; cristhco@ucm.es (C.A.C.-G.); luis.puente@salud.madrid.org (L.P.-M.); javier.miguel@salud.madrid.org (J.d.M.-D.); 2Faculty of Medicine, Complutense University of Madrid, 28040 Madrid, Spain; javierpl@ucm.es (J.P.-H.); igorroch@ucm.es (I.G.-M.); 3Gregorio Marañón Biomedical Research Institute, 28007 Madrid, Spain; 4Respiratory Department, Infanta Leonor University Hospital, 28031 Madrid, Spain; irenemilagros.dominguez@salud.madrid.org; 5Department of Public Health and Maternal & Child Health, Faculty of Medicine, Universidad Complutense de Madrid, 28040 Madrid, Spain; anailo04@ucm.es (A.L.-d.-A.); rodrijim@ucm.es (R.J.-G.); josejzam@ucm.es (J.J.Z.-L.)

**Keywords:** chronic obstructive pulmonary disease, quality of life, disease exacerbation, symptom cluster

## Abstract

Background: Chronic obstructive pulmonary disease (COPD) exacerbations represent significant clinical events marked by worsening respiratory symptoms, often necessitating changes in medication or hospitalization. Identifying patterns of exacerbation and understanding their clinical implications are critical for improving patient outcomes. This study aimed to identify exacerbation patterns in COPD patients using variations in the COPD Assessment Test (CAT) scores and compare clinical characteristics and comorbidities among patients with different exacerbation patterns. Methods: An observational study was conducted involving COPD patients admitted for severe exacerbations. The administered CAT questionnaire referred to two periods: (1) the period during hospital admission and (2) the stable period two months prior to admission. Results: Fifty patients (60% male, mean age 70.5 years, standard deviation [SD] 9.6) were included; of these, eight (16%) were active smokers. Significant worsening in CAT scores during the exacerbation compared to the stable period was observed (25 vs. 13.5, *p* < 0.001). Three exacerbation patterns were identified: increased cough and sputum (cluster 1); increased dyspnea and activity limitation (cluster 2); and poorer sleep quality and lower energy (cluster 3). No significant differences were found regarding demographics and lung function. Conclusions: Three distinct exacerbation patterns were identified in COPD patients based on CAT score variations, suggesting that exacerbations are heterogeneous events. Future studies with larger sample sizes and prospective follow-up are necessary to validate these findings and explore their clinical and prognostic implications.

## 1. Introduction

Chronic obstructive pulmonary disease (COPD) is a complex disorder. Over 80% of patients with COPD have at least one additional chronic condition, including cardiovascular diseases, diabetes, and psychological disorders, which in turn are associated with a higher risk of mortality [1,2]. Identifying and understanding comorbidities is crucial for improving clinical care for these patients, establishing research priorities, and developing clinical practice guidelines [3].

The pathophysiology of COPD involves a combination of airway inflammation, oxidative stress, and structural changes in the lungs [1,2]. Inflammation in COPD is driven by the recruitment of inflammatory cells and the release of proinflammatory mediators, leading to tissue damage and airway remodeling [1]. The structural changes in COPD include airway obstruction, mucus hypersecretion, and destruction of alveolar walls, resulting in airflow limitation and reduced gas exchange [1].

Another fundamental aspect of COPD is exacerbations. These represent acute events characterized by worsening respiratory symptoms beyond normal day-to-day variations, leading to medication changes and potentially requiring hospitalization [4].

Furthermore, exacerbations in COPD can have lasting effects on lung function, airway inflammation, and future susceptibility to these events [5]. Patients who experience exacerbations may exhibit increased airway inflammation, accelerated decline in lung function, and greater bacterial colonization, leading to a higher risk of recurrence [5].

Identifying patients at risk of frequent exacerbations, understanding the different types, and knowing their long-term consequences are crucial aspects for implementing interventions aimed at preventing and managing these events, optimizing COPD management, improving patient outcomes, and reducing disease burden [5].

In routine clinical practice, validated questionnaires, such as the COPD Assessment Test (CAT), are available to assess the impact of the disease on patients’ health status. Some studies have shown that a score of ≥15 on this questionnaire can predict the risk of exacerbations [5,6].

The CAT questionnaire consists of a set of eight questions related to the symptoms and quality of life of patients with COPD. Each item is scored between 0 and 5, where 0 represents the best health status and 5 the worst. The total score can range from 0 to 40, with a higher score indicating a greater impact of COPD on the patient’s quality of life.

COPD exacerbations are not uniform events. Each can vary in severity, impact, and underlying triggers. Therefore, there are different types of exacerbations that carry variable risks and implications for patients [7]. For example, exacerbations triggered by viral respiratory infections may present differently from those caused by bacterial infections, with distinct clinical manifestations and treatment responses [8].

The objectives of this study were: (1) to identify patterns of symptoms associated with exacerbation through variations in the CAT questionnaire score and (2) to compare clinical characteristics and the presence of comorbidities among patients according to these exacerbation patterns.

## 2. Materials and Methods

### 2.1. Design and Study Population

An observational, non-interventional study was conducted on patients with COPD admitted for a severe exacerbation between September 2021 and March 2023. Inclusion criteria were: (1) a previously established diagnosis of COPD based on post-bronchodilator spirometry findings with an FEV1/FVC ratio below 0.70; (2) being over 40 years of age at the time of study inclusion; and (3) being able to understand and sign the informed consent document. Exclusion criteria included: (1) cognitive impairment or other alterations that might hinder the completion of the CAT questionnaire; (2) having a COPD exacerbation within the two months prior to study inclusion; and (3) a history of ICU admission due to a COPD exacerbation. Due to the high clinical workload during the recruitment period, participation was not offered to all patients who met the selection criteria. Participation in the study was offered within the constraints of our workload, and patients meeting the inclusion criteria were selected randomly, without considering any other factors, in order to minimize selection bias as much as possible.

### 2.2. Collected Variables

All variables were collected at the time of study inclusion. Medical histories and anthropometric and clinical data at the start of the hospital admission were recorded. The latest available spirometry findings before hospital admission were also reviewed. Participants completed the CAT questionnaire twice at the same study visit: once at the start of hospital admission and once in relation to the stable period two months before admission.

### 2.3. Statistical Analysis

Qualitative variables were expressed as frequencies and percentages. Quantitative variables with normal distribution were represented as the mean and standard deviation (SD). Quantitative variables that were not normally distributed were shown as the median and interquartile range (IQR). Normal distribution was determined by histogram analysis. Clinical and functional characteristics were compared using Student’s t-test for quantitative variables and Fisher’s exact test for qualitative variables. The two CAT questionnaire scores were considered repeated measures, and their comparison was conducted using Friedrich’s test.

A discriminant analysis was performed to identify worsening patterns in the CAT questionnaire. For this analysis, a hierarchical cluster analysis was performed using the average linkage method based on standardized squared Euclidean distance [9,10] to group subjects into three clusters, considering the eight items in the CAT questionnaire. No additional data standardization was applied. A proximity matrix was generated to calculate distances between subjects, and the results were visualized through a dendrogram [11,12].

A bilateral *p*-value of less than 0.05 was considered statistically significant for all comparisons. All statistical analyses were conducted using SPSS version 26 (IBM Corp., Armonk, NY, USA).

### 2.4. Ethical Aspects

This study was approved by the Research Ethics Committee with Medications of the General University Hospital Gregorio Marañón with code 14/2019 and date of approval 21 October 2019. Participants provided written informed consent.

## 3. Results

Fifty patients were included in the study, with a mean age of 70.5 years (SD 9.6). Of these, 30 (60%) were male, and 8 (16%) were active smokers. The mean values for weight, height, and body mass index (BMI) were 72.2 kg (SD 15.9), 163.6 cm (SD 7.8), and 27.0 kg/m^2^ (SD 5.6), respectively.

Regarding lung function, the mean absolute FEV1 value was 1.17 L (SD 0.46) and 46.7% (SD 8.0) when expressed as a percentage of the predicted value (FEV1pp). The mean absolute FVC value was 2.47 L (SD 0.84) and 75.7% (SD 18.4) as a percentage of the predicted value (FVCpp). The total and item-specific scores of the CAT questionnaire, both at admission and during the stable period, as well as their comparisons, are shown in Table 1.

Figure 1 illustrates the dendrogram of COPD exacerbation patterns based on CAT score variations.

Three patterns of worsening in the CAT questionnaire scores were identified: increased cough and sputum (cluster 1); increased dyspnea and limitation for activities (cluster 2); and poorer sleep quality and lower energy (cluster 3). Clinical characteristics and comorbidities by cluster are shown in Table 2 and Table 3, respectively.

## 4. Discussion

In the present study, both the total score and the score of each item on the CAT questionnaire significantly worsened during acute exacerbation episodes of COPD compared to the baseline CAT score. The results indicate that during exacerbations, patients experience a notable increase in respiratory symptoms, including dyspnea and cough, as well as greater limitations in daily activities and a general decrease in quality of life. Additionally, three different clusters of worsening symptoms reported in the CAT during severe exacerbations were identified, with the most frequent being patients who presented with increased cough and sputum (cluster 1), compared to those with dyspnea associated with activity limitation (cluster 2) and those reporting poorer sleep quality associated with less energy (cluster 3).

Patients in cluster 1 likely experienced a COPD exacerbation due to a respiratory infection, while patients in cluster 3 probably had greater asthenia due to the presence of other comorbidities that could produce similar symptoms and may have worsened during the COPD exacerbation. However, no clear explanation was found for the differences observed in the patients from cluster 2. It is possible that pre-existing differences in the characteristics of COPD between patients in clusters 1 and 2 during the stable phase could account for this, such as a higher prevalence of chronic bronchitis in cluster 1 patients and a greater extent of emphysema in cluster 2 patients. Nonetheless, this hypothesis needs to be validated in larger, targeted future studies.

Patients from different groups were not distinguishable by their baseline clinical characteristics. Regarding the presence of comorbidities, a higher prevalence of diabetes mellitus, chronic kidney disease, and solid tumors without metastasis was documented in patients from cluster 3.

The clusters identified in this study were based exclusively on the variations in CAT score. However, no data were collected regarding cellular or molecular biomarkers such as eosinophils or C-reactive proteins. These biomarkers could give deeper insights into the pathophysiology behind these patterns. Previous studies have indicated that these biomarkers might be linked to various exacerbation patterns and therefore, affect therapeutic treatment choice as well as prognosis of patients [13,14]. Thus, there is need for future studies to consider adding biomarker analysis to better understand the molecular and cellular disparity between exacerbation patterns.

Although the CAT questionnaire is generally administered during the stable phase, some authors have used this questionnaire to monitor treatment response during a COPD exacerbation [15]. A decrease of 3.5 points has been used as a cutoff to differentiate responders from non-responders to treatment [15]. In our study, although the CAT questionnaire was used during the exacerbation phase, the purpose was not to monitor improvement with treatment but to assess the magnitude of worsening of different aspects at the onset of exacerbation compared to the stable phase.

This offers a unique perspective on how the CAT questionnaire can detect fluctuations in patient symptoms at different stages of the disease, beyond its usual application during the stable phase. Retrospective assessment using the CAT questionnaire during an exacerbation can provide valuable insights into which specific domains (such as dyspnea, cough, or limitation in daily activities) are most affected in each patient, which in turn could influence the personalization of treatment.

There have been previous attempts in the literature to distinguish different forms of COPD exacerbation based on pathophysiological mechanisms, exacerbation etiology, or severity [8,16]. To our knowledge, our study is the first to group these patients by worsening symptoms and the impact on their quality of life.

In clinical practice, classifying similar patients to apply more individualized treatment is an increasingly common strategy. For example, the Spanish COPD management guidelines (GesEPOC) establish different phenotypes in patients with stable COPD [17]. Moreover, studies have established decompensation patterns in other diseases, such as heart failure [18] and chronic kidney disease [19].

The identification of different COPD exacerbation patterns can enable healthcare professionals to individualize treatment strategies, aiming to address the most prevalent symptoms, triggers, and specific risk factors associated with each pattern. This could lead to more effective management and better patient outcomes [20,21]. A deeper understanding of exacerbation patterns could also enable clinicians to anticipate disease progression and adjust maintenance therapies before a severe exacerbation occurs. Additionally, it could promote greater treatment adherence among patients, as strategies would be more closely aligned with their specific experiences and needs, potentially reducing hospitalization rates and associated mortality.

Our study did not find any clinical or functional characteristics that were specific to each exacerbation pattern during the patient’s stable phase. This is likely due to the limited statistical power resulting from the small sample size. However, in relation to the presence of comorbidities, patients in cluster 3 had a higher prevalence of diabetes mellitus, chronic kidney disease, and solid tumors without metastasis. All these comorbidities are known to exacerbate systemic inflammation and oxidative stress, which may worsen COPD symptoms and impair sleep quality, and the chronic inflammation and metabolic dysregulation may lead to fatigue and asthenia [22,23,24,25]. Thus, COPD exacerbation in these patients could lead to worsening control of these comorbidities, which should be considered in the comprehensive management of their disease. Due to the limited number of patients in the cluster 3, these observations should be interpreted with caution and not considered conclusive.

This finding underscores the importance of a multidisciplinary approach in the treatment of COPD patients, particularly those with multiple comorbidities. Focusing solely on lung function or respiratory symptoms is insufficient; it is also crucial to optimize the management of comorbidities to prevent them from exacerbating the course of COPD, or vice versa. Therefore, individualized treatment should include a comprehensive assessment of comorbidities and their potential impact during exacerbations.

Our study has a number of limitations. The main limitation of this study is the small sample size, both in the overall study and specifically in cluster 3, which resulted from significant disruptions to patient recruitment due to the peak care burden during the COVID-19 pandemic. Initially, the study was designed in 2019 with the aim of enrolling a larger cohort of patients with COPD to achieve greater statistical power for identifying exacerbation patterns through variations in CAT scores. However, the unforeseen challenges brought on by the pandemic, including prolonged interruptions in patient inclusion, limited our ability to reach the intended sample size. Consequently, some clusters contained a small number of patients, thereby reducing the statistical power of our findings. Nevertheless, the insights gained from this smaller cohort offer valuable preliminary data, which will help us to design more robust studies in the future. Moreover, this study does not include prospective follow-up of the included patients, so it is unknown whether the prognosis of patients in each cluster differs. Thirdly, the retrospective administration of the CAT questionnaire in relation to the period two months before the exacerbation could introduce recall bias. Taking recall bias into account and to facilitate study procedures during a high-demand care period, we designed this study to be a single-visit study. Due to this potential recall bias, caution must be exercised when interpreting the results of this study, and the findings of our study should be validated in future studies with a prospective design. Being a single-center study, the sample may not represent the general population characteristics of COPD patients. Another limitation is the absence of multivariate analysis. This is due to the fact that the univariate analysis identified few variables with significant differences of clinical interest. Therefore, multivariate analysis was not performed, as the limited statistical power would not allow for the generation of convincing results.

Another aspect that warrants consideration is the intrinsic heterogeneity of COPD. The disease is characterized by variability in clinical presentation and treatment response, which complicates the standardization of management and the interpretation of results in studies involving diverse populations. In this context, identifying specific exacerbation patterns through the CAT questionnaire may be a first step toward addressing COPD from a more personalized perspective. However, it also highlights the need to develop more refined and differentiated assessment tools that can capture the unique characteristics of each patient.

Considering these limitations, a larger study with more patients is necessary to achieve higher statistical power to detect differences in the clinical characteristics of patients in different clusters. Nonetheless, our study demonstrates the possibility of identifying different COPD exacerbation patterns based on the most affected symptoms and quality of life.

In future studies, the integration of specific biomarkers, both inflammatory and genetic, could provide additional insights into the underlying mechanisms of different exacerbation patterns. Correlating these clinical patterns with specific biomarkers could enhance our understanding of the pathophysiology of exacerbations and pave the way for more targeted therapeutic approaches.

Additionally, the psychological impact of exacerbations in COPD patients should also be considered. The worsening of dyspnea and limitation in daily activities can lead to increased anxiety and depression, which in turn may exacerbate respiratory symptoms and worsen disease control. Therefore, incorporating psychological assessments in future studies could provide a more comprehensive understanding of the impact of exacerbations and help develop intervention strategies that address both the physical and mental aspects of COPD.

## 5. Conclusions

In the present study, three patterns of COPD exacerbation were identified based on worsening symptoms and quality of life measured by the CAT questionnaire. Due to limited statistical power, no statistically significant differences were observed regarding clinical and functional characteristics, but a higher prevalence of diabetes mellitus, chronic kidney disease, and solid tumors without metastasis was documented in patients from cluster 3 as a preliminary result of this study with an exploratory nature. This suggests that there are subgroups of COPD patients in whom the presence of certain comorbidities may have important implications during severe exacerbations. A larger study, including prospective follow-up and taking into account other aspects of COPD, is needed to identify clinical and prognostic differences among patients in different clusters.

## Figures and Tables

**Figure 1 diseases-12-00283-f001:**
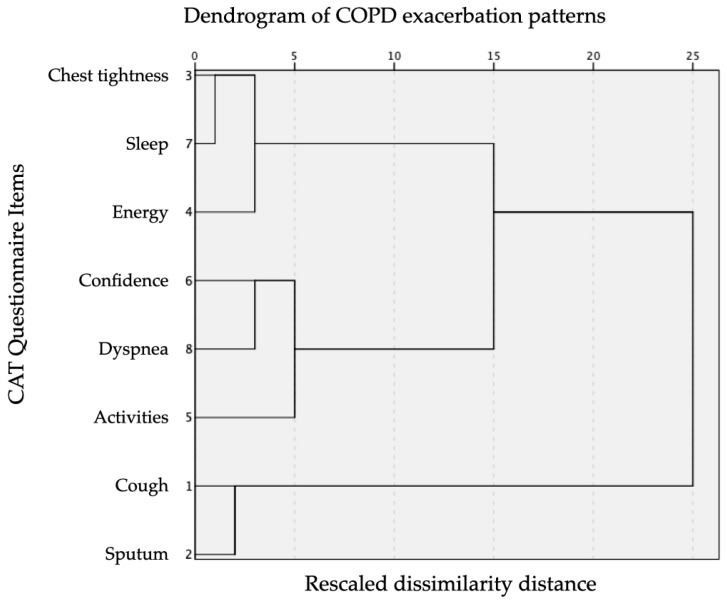
Dendrogram of COPD exacerbation patterns based on CAT score variations.

**Table 1 diseases-12-00283-t001:** Comparison between baseline CAT score and CAT score during exacerbation.

Item, Median (IQR)	Baseline CAT	CAT During Exacerbation	Difference	*p*-Value
Cough	1 (0–3)	3 (2–4)	1 (0–2)	<0.001
Sputum	1.5 (0–3)	3.5 (1.75–4.25)	1 (0–2)	<0.001
Chest tightness	0 (0–1.25)	2 (0–4)	0 (0–2)	<0.001
Dyspnea	3 (2–5)	5 (4–5)	1 (0–2)	<0.001
Activities	1.5 (0–3)	4 (1–5)	2 (0–3)	<0.001
Confidence	0.5 (0–2.25)	3.5 (0.75–5)	1.5 (0–3)	<0.001
Sleep	0 (0–1.25)	1 (0–4)	0 (0–2)	<0.001
Energy	2 (0–3)	3.5 (2.75–5)	2 (0–2)	<0.001
Total	13.5 (7–19)	25 (17.5–30)	9 (5–15.25)	<0.001

Abbreviations: CAT: COPD Assessment Test; IQR: Interquartile Range.

**Table 2 diseases-12-00283-t002:** Comparison of clinical characteristics and Charlson Index score by exacerbation cluster.

Characteristic	Cluster 1 (n = 34)	Cluster 2 (n = 11)	Cluster 3 (n = 5)	*p*-Value
Male, n (%)	21 (61.8)	6 (54.5)	3 (60.0)	0.914
Age, years (SD)	72 (8)	68 (13)	68 (10)	0.499
Weight, kg (SD)	72 (17)	74 (13)	71 (17)	0.905
Height, m (SD)	1.64 (0.07)	1.63 (0.10)	1.61 (0.10)	0.679
FEV1, L (SD)		1.20 (0.50)	1.11 (0.39)	1.08 (0.24)
FEV1pp, % (SD)	47 (17)	47 (17)	42 (7)	0.787

Abbreviations—FEV1: forced expiratory volume in the first second; FEV1pp: FEV1 as a percentage of the predicted value.

**Table 3 diseases-12-00283-t003:** Comparison of comorbidities and Charlson Index score by exacerbation cluster.

Comorbidity	Cluster 1	Cluster 2	Cluster 3	*p*-Value
Arterial hypertension, n (%)	17 (51.5)	7 (63.6)	4 (80.0)	0.431
Dyslipidemia, n (%)	13 (39.4)	7 (63.6)	4 (80.0)	0.130
Diabetes mellitus, n (%)	6 (18.2)	2 (18.2)	4 (80.0)	0.010
Atrial fibrillation, n (%)	4 (12.1)	1 (9.1)	0 (0.0)	0.699
Ischemic heart disease, n (%)	4 (12.1)	0 (0.0)	2 (40.0)	0.077
Heart failure, n (%)	6 (18.2)	2 (18.2)	2 (40.0)	0.518
Peripheral vascular disease, n (%)	0 (0.0)	0 (0.0)	0 (0.0)	-
Cerebrovascular disease, n (%)	3 (9.1)	1 (9.1)	0 (0.0)	0.781
Sleep apnea, n (%)	6 (18.2)	2 (20.0)	0 (0.0)	0.567
Dementia, n (%)	0 (0.0)	0 (0.0)	0 (0.0)	-
Connective tissue disease, n (%)	0 (0.0)	1 (9.1)	0 (0.0)	0.171
Gastric ulcer, n (%)	0 (0.0)	0 (0.0)	0 (0.0)	-
Liver disease, n (%)	3 (9.1)	2 (18.2)	0 (0.0)	0.502
Hemiplegia, n (%)	0 (0.0)	0 (0.0)	0 (0.0)	-
Chronic kidney disease, n (%)	2 (6.1)	0 (0.0)	2 (40.0)	0.019
Solid tumor without metastasis, n (%)	3 (9.1)	1 (9.1)	3 (60.0)	0.009
Leukemia, n (%)	1 (3.0)	0 (0.0)	0 (0.0)	0.781
Lymphoma, n (%)	0 (0.0)	0 (0.0)	0 (0.0)	-
Solid tumor with metastasis, n (%)	0 (0.0)	0 (0.0)	0 (0.0)	-
HIV, n (%)	0 (0.0)	0 (0.0)	0 (0.0)	-
Anxiety, n (%)	2 (6.1)	0 (0.0)	0 (0.0)	0.603
Depression, n (%)	3 (9.1)	2 (18.2)	1 (20.0)	0.623
Idiopathic pulmonary fibrosis, n (%)	0 (0.0)	0 (0.0)	0 (0.0)	-
Cirrhosis, n (%)	1 (3.0)	0 (0.0)	1 (20.0)	0.150
Charlson Index, median (interquartile range)	2 (1–4)	2 (1–3)	7 (1.5–9)	0.132

Abbreviations—SD: standard deviation.

## Data Availability

The data that support the findings of this study are available from the corresponding author upon reasonable request.

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
