# Peer review of "Deterioration Patterns in Patients Admitted for Severe COPD Exacerbation"

_diseases, 2024, doi:10.3390/diseases12110283_

Round 1

Reviewer 1 Report

Comments and Suggestions for Authors

The authors present a manuscript in which they identify three clinical patterns of COPD exacerbation to enable early diagnosis.

Comments

1) Cigarette smoking habit should be added to the demographic characteristics of patients.
2) The presence of multiple independent variables should be analyzed by multivariate analysis. The authors should comment on this point
3) The authors list the limitations of the study.  They should add to the limitations the retrospective nature of the study and the need for future studies that are not only larger in sample size but also prospective and possibly randomized and controlled.
4) The authors should formulate hypotheses on the pathophysiological mechanism related to sleep alterations and lack of energy (cluster 3) in the predictivity of CODP exacerbation.

Author Response

Comment 1: Cigarette smoking habit should be added to the demographic characteristics of patients.

Response 1: First, I would like to thank the reviewer for their efforts in improving the quality of the manuscript. Smoking habits are described in the demographic data, specifically in the first paragraph of the Results section: “Of these, 30 (60%) were male, and 8 (16%) were active smokers”. Smoking status is recorded as a dichotomous variable: active smoking Y/N. All patients who were not active smokers were former smokers.

Comment 2: The presence of multiple independent variables should be analyzed by multivariate analysis. The authors should comment on this point.

Response 2: Taking into account the Reviewer’s comment, we have added a sentence in the Limitations section.

Another limitation is the absence of a multivariate analysis. This is due to the fact that the univariate analysis identified few variables with significant differences of clinical interest. Therefore, a multivariate analysis was not performed, as the limited statistical power would not allow for the generation of convincing results.

Comment 3: The authors list the limitations of the study.  They should add to the limitations the retrospective nature of the study and the need for future studies that are not only larger in sample size but also prospective and possibly randomized and controlled.

Response 3: We appreciate the Reviewer’s comment. The descriptive nature and the retrospective administration of the CAT questionnaire are described in the Limitations section: “Thirdly, the retrospective administration of the CAT questionnaire in relation to the period two months before the exacerbation could introduce recall bias”. We have added the need for future prospective studies to verify our findings.

Due to this potential recall bias, caution must be exercised when interpreting the results of this study, and the findings of our study should be validated in future studies with a prospective design.

Comment 4: The authors should formulate hypotheses on the pathophysiological mechanism related to sleep alterations and lack of energy (cluster 3) in the predictivity of CODP exacerbation.

Response 4: We appreciate the Reviewer’s suggestion. In response, we hypothesize that the increased presence of comorbidities such as diabetes mellitus, chronic kidney disease, and solid tumors in this cluster may contribute to the pathophysiological mechanisms underlying sleep disturbances and lack of energy. These conditions are known to exacerbate systemic inflammation and oxidative stress, which may worsen COPD symptoms and impair sleep quality, and the chronic inflammation and metabolic dysregulation may lead to fatigue and diminished energy levels.

All these comorbidities are known to exacerbate systemic inflammation and oxidative stress, which may worsen COPD symptoms and impair sleep quality, and the chronic inflammation and metabolic dysregulation may lead to fatigue and asthenia(22–25).

Reviewer 2 Report

Comments and Suggestions for Authors

This manuscript reports patterns found in the deterioration of patients admitted to hospital for severe chronic obstructive pulmonary disease (COPD) based upon a sample of patients who undertook the COPD assessment test in relation to both their condition at time of admission and retrospectively concerning their health prior to admission. By the authors’ own admission, this study has a limited sample size to the extent that the conclusions drawn should be considered preliminary in nature. However, I feel that the study offers a useful initial framework for assessing the deterioration of COPD patients and makes an initial attempt at identifying salient comorbidities that are differentiated across clusters. It is understood that the results presented will need to be built upon by future studies. As such, I think that the paper offers some value for readers of ‘Diseases’. There are some points that the authors could address, and these are detailed below.

1.      Page 2 lines 89-90 “Due to the . . . the selection criteria.” Whilst I appreciate the pressures under which this study was conducted, given you were not offering trial participation to all eligible patients, was there an objective set of criteria used to select which patients it was offered to, on and above the eligibility criteria identified in section 2.1? Can we be assured that selection took place without any undue bias on the sample set included?

2.      Page 2 line 96 “. . . in relation to . . . before admission.” Was the retrospective questionnaire completed at the same time as the admission questionnaire? I am assuming it must have been, given you later refer to this being all done in a single session, but it is worthwhile clarifying this at this point in the manuscript.

3.      Table 1. This contains two categories where the difference between baseline and during exacerbation is zero (chest tightness and sleep) yet these have been given p-values of <0.001. Given that there is clearly no difference in these cases, why do the p-values indicate significance of difference? Some clarification is needed here.

4.      Page 7 lines 261-263 “For example . .  and emphysema.” Do you really think that you have sufficient data to conclude statistically significant differences like this between the clusters? Provide a better connection between these statements and the data presented in the paper.

Comments on the Quality of English Language

The quality of the English in this manuscript is acceptable. A minor editorial check may be advisable.

Author Response

Comment 1: Page 2 lines 89-90 “Due to the . . . the selection criteria.” Whilst I appreciate the pressures under which this study was conducted, given you were not offering trial participation to all eligible patients, was there an objective set of criteria used to select which patients it was offered to, on and above the eligibility criteria identified in section 2.1? Can we be assured that selection took place without any undue bias on the sample set included?

Response 1: First, I would like to thank to the Reviewer for the positive feedback on our study and for the effort to improve the quality of this manuscript. Regarding the Reviewer’s comment, I would like to assure that participation in the study was offered within the constraints of our workload, and patients meeting the inclusion criteria were selected randomly, without considering any other factors, in order to minimize selection bias as much as possible. We added this description to the Methods section of the revised version of the manuscript.

The participation in the study was offered within the constraints of our workload, and patients meeting the inclusion criteria were selected randomly, without considering any other factors, in order to minimize selection bias as much as possible.

Comment 2: Page 2 line 96 “. . . in relation to . . . before admission.” Was the retrospective questionnaire completed at the same time as the admission questionnaire? I am assuming it must have been, given you later refer to this being all done in a single session, but it is worthwhile clarifying this at this point in the manuscript.

Response 2: The patients completed the CAT questionnaire twice at the same time. We have added this description in the revised version of the manuscript to avoid any confusion.

Participants completed the CAT questionnaire twice at the same study visit: once at the start of hospital admission and once in relation to the stable period two months before admission.

Comment 3: Table 1. This contains two categories where the difference between baseline and during exacerbation is zero (chest tightness and sleep) yet these have been given p-values of <0.001. Given that there is clearly no difference in these cases, why do the p-values indicate significance of difference? Some clarification is needed here.

Response 3: We appreciate the Reviewer’s comment. The value of 0 refers to the median difference between the two CAT questionnaire measurements. Although the median for the item during exacerbation and the stable phase coincidentally match, there are differences in the other subjects. This is evident from the different interquartile ranges of the item during exacerbation and in the stable phase. The p-Value results from a non-parametric comparison of both measurements.

Comment 4: Page 7 lines 261-263 “For example . .  and emphysema.” Do you really think that you have sufficient data to conclude statistically significant differences like this between the clusters? Provide a better connection between these statements and the data presented in the paper.

Response 4: We understand that the wording of the manuscript should be more rigorous. This paragraph reflects our intention to include inflammatory markers in future studies, but it is true that the example lacks a solid scientific basis. Therefore, we have decided to remove this example in the revised version of the manuscript.

Reviewer 3 Report

Comments and Suggestions for Authors

See the attached review report.

Author Response

  1. Abstract:

Comment A. Rather than giving methods in the abstracts, summarize the results.

Response A: In accordance with the journal's instructions for authors, both methods and results must be included in the abstract. Additionally, there is a word limit for the Abstract section. Taking into account the Reviewer’s comment, we have slightly expanded the description of the results in the abstract as much as possible.

Comment B: Give values.

Response B: We have expanded the numerical data as much as possible.

Comment C: Emphasize on the key factor responsible for COPD progressing.

Response C: We appreciate the Reviewer’s comment. However, our article does not study the progression of COPD, but rather the patterns of deterioration during a COPD exacerbation. Our study is not prospective, a fact discussed in the Limitations section. To study disease progression, it would be necessary to observe clinical and functional evolution after the COPD exacerbation, which is not the aim of this study. Therefore, we do not have data on this aspect.

  1. Introduction:

Comment A: How would sever COVID lead to failure of other exams?

Response A: Dear Reviewer, our article does not study the relationship between COPD and COVID, nor did we conduct any examinations related to COVID, so we are unsure what you are referring to by 'failure of other exams.' The only mention of COVID in the article is the limitation due to the high workload caused by the pandemic situation while we were recruiting patients for this study, which is already discussed in the manuscript, but it is not directly related to COVID itself.

Comment B: Elaborate on predictions using questionnaires.

Response B: Dear Reviewer, our study does not aim to make predictions using questionnaires, but rather to identify different patterns of deterioration through questionnaires. Our study is not prospective. To make predictions, prospective data would be required, but that is not the objective of this study.

Comment C: Explain more about CAT questionnaire score.

Response C: We added the following paragraph.

The CAT questionnaire consists of a set of 8 questions related to the symptoms and quality of life of patients with COPD. Each item is scored between 0 and 5, where 0 represents the best health status and 5 the worst. The total score can range from 0 to 40, with a higher score indicating a greater impact of COPD on the patient’s quality of life.

Comment D: The authors should highlight the research gap and aim of the study.

Response D: The background of the study is described in the penultimate paragraph of the Introduction section: COPD exacerbations are not uniform events. The objectives are outlined in the final paragraph of the Introduction section, as follows: The objectives of this study were: 1) to identify patterns of symptoms associated with exacerbation through variations in the CAT questionnaire score; and 2) to compare clinical characteristics and the presence of comorbidities among patients according to these exacerbation patterns. Both are already described concisely.

  1. Methods:

Comment A: What was the reason for including completion of the CAT questionnaire?

Response A: It was the primary objective of this study.

Comment B: The results include BMI, what was the inclusion BMI criteria considered?

Response B: BMI was one of the multiple variables collected, but it was not an inclusion criteria.

Comment C: The same goes for smoking.

Response C: The same explanation as BMI.

Comment D: Define the age range considered.

Response D: Dear Reviewer, the answer to your comment is already described in the Methods section of the manuscript, specifically in inclusion criterion number 2: Being over 40 years of age at the time of study inclusion.

Comment E: Explain methodology in detail.

Response E: Dear Reviewer, the methodology is already described in detail. In fact, the Methods section is subdivided into four parts: Design and study population, collected variables, statistical analysis, and ethical considerations. If any information seems to be missing, it might be found upon rereading, as occurred with your previous comment. If any detail is still lacking, please specify it.

  1. Results:

Comment A: How did the authors calculate Charlson index?

Response A: Recording the presence of each comorbidity included in the Charlson index, multiplying each comorbidity by its weighting factor, and summing the results.

  1. Discussion:

Comment A: Interrelate results!

Response A: We have interrrelated our findings to the existing literature. In fact, we have cited 13 references for this purpose.

Comment B: Data visualizations can be improved.

Response B: The majority of the data are presented either in tables or in the figure.

Comment C: May be other factors like diet, exercise would effect COPD.

Response C: Unfortunately, diet and exercise are not part of the objectives of our study, so we do not have data to discuss these aspects. The main objective of the study is to identify different patterns of deterioration during a COPD exacerbation using the CAT questionnaire.

Comment D: The results should be interrelated.

Response D: We have interrrelated our findings to the existing literature. In fact, we have cited 13 references for this purpose.

Round 2

Reviewer 1 Report

Comments and Suggestions for Authors

The questions have been answered satisfacorily